# Combining Biocontrol Agents with Chemical Fungicides for Integrated Plant Fungal Disease Control

**DOI:** 10.3390/microorganisms8121930

**Published:** 2020-12-04

**Authors:** Lena Ons, Dany Bylemans, Karin Thevissen, Bruno P. A. Cammue

**Affiliations:** 1Centre for Microbial and Plant Genetics, KU Leuven, Kasteelpark Arenberg 20, 3001 Heverlee, Belgium; lena.ons@kuleuven.be (L.O.); karin.thevissen@kuleuven.be (K.T.); 2Department of Biosystems, KU Leuven, Decroylaan 42, 3001 Heverlee, Belgium; dany.bylemans@kuleuven.be; 3Research Station for Fruit, Fruittuinweg 1, 3800 Sint-Truiden, Belgium

**Keywords:** fungicide combinations, integrated pest management, biocontrol, induced resistance, antagonism

## Abstract

Feeding a rising population of currently 7.8 billion people globally requires efficient agriculture, which is preferably sustainable. Today, farmers are largely dependent on synthetic fungicides to avoid food losses caused by fungal diseases. However, the extensive use of these has resulted in the emergence of fungicide-resistant pathogens and concerns have been raised over the residual effects on the environment and human health. In this regard, biocontrol agents (BCAs) have been proposed as an alternative to standard fungicides but their disease management capacity is usually incomplete and heavily relies on uncontrollable environmental conditions. An integrated approach combining BCAs with fungicides, which is the focus of this review, is put forward as a way to reduce the fungicide doses to manage plant diseases and thereby their residue on harvested crops. In addition, such a strategy of combining antifungal treatments with different modes of action reduces the selection pressure on pathogens and thereby the chances of resistance development. However, to allow its large-scale implementation, further knowledge is needed, comprising timing, number and interval of repeated BCA applications and their compatibility with fungicides. The compatibility of BCAs with fungicides might differ when applied in a mixture or when used in alternation.

## 1. Introduction

Plant diseases and pests are a major threat to global food availability. For example, the potential food losses due to diseases, caused by pathogenic micro-organisms and animal pests, are estimated to be up to 38.2% of total yield losses in rice and 36.5% in potatoes [1]. According to the United Nations, the world population is expected to increase to 9.7 billion by 2050, which means that a dramatic increase in global crop harvest is required in order to satisfy the population’s food needs [2]. This can be realized by increasing the area of cropland, disrupting natural ecosystems or by intensifying crop yields [3]. However, crops with increased yield are often associated with even higher vulnerability to diseases and pests. In particular, fungal plant pathogens are attracted to nutrient-dense plant tissues. As such, the potential loss of wheat production due to fungal diseases increases from less than 10%, with an attainable yield of 2 tons/ha, to more than 20% when the intensity of production increases to 12 tons/ha [1]. The use of synthetic pesticides has therefore become an integral part of agriculture. As such, since the discovery of the first synthetic fungicide, phenylmercury acetate in 1913, over 110 new fungicides have been developed during the last century, allowing food production to increase with a value of USD 12.8 billion in the US annually [4,5,6] However, their extensive use has encountered two main challenges. First, concerns have been raised over the residual effects and toxicity that affect the environment and human health. For example, fungicides, and other types of pesticides, have recently been linked to cancer and respiratory and hormone imbalance diseases, thereby depending on the level of exposure [7,8,9]. Driven by the opinion of consumers, who perceive pesticides as a threat, and the vast amount of research supporting this view, regulators have approved laws that result in either banning or restricting their use by imposing lower maximum residue limits (MRLs) [10,11]. In the European Union (EU), the MRL review program was implemented under Regulation 396/2005 to restrict the use of synthetic pesticides. Second, the efficacy of fungicides has decreased due to the emergence of resistant pathogens [12]. However, the discovery of new types of fungicides has become more difficult and more costly [13]. As such, the cost of the discovery and development of one new active ingredient increased from USD 195 million in 1995 to USD 286 million in 2016 [13].

In response to the increasing knowledge about the negative side effects of pesticide overuse, integrated pest management (IPM) was implemented. IPM is defined as the best mix of plant disease control strategies, taking into account the crop yield, profit and safety profile, as presented by the Food and Agriculture Organization of the United Nations (FAO) [14]. Worldwide, IPM is an accepted strategy to reduce pesticide usage in pest management [15]. In the EU, the sustainable application of pesticides is required by directive 2009/128 [16]. Particular emphasis is placed on the prevention of infection and the consideration of all available plant disease management tools while taking into account their economic benefits and toxicity. In this regard, biocontrol has been proposed as an alternative to conventional pesticides.

The term biological control or biocontrol often causes confusion as different meanings circulate in the scientific literature. In the most narrow definition, biological control can be defined as the use of a living organism to act against a specific plant pathogen or pest via parasitism, antibiosis or competition for nutrients or space [17]. However, plant diseases and pests are induced and regulated by complex processes on different levels: the invader—being the plant pathogen or pest—the environment and the plant itself. They only thrive if conditions are optimal on all three levels [18]. Therefore, a broader definition of biological control, covering all levels, is needed to in order to achieve its true potential in disease and pest management. This broader term includes the application of living organisms and their derivatives to control plant diseases and pests, not only via direct antagonistic effects against plant pathogens and pests but also indirectly via the induction of resistance [19]. The differences with the narrow definition of biocontrol are therefore that derivatives of living organisms and inducers of resistance, which activate the defense mechanisms of plants, are also defined as biocontrol agents (BCAs) [20]. Examples include parasitoid wasps, predatory mites against several pests such as potato tuber moth and pathogenic bacteria and fungi, like *Bacillus* spp. and *Trichoderma* spp., which act against different types of plant pathogens [21,22,23,24]. Along with biocontrol organisms, there are BCAs such as chitosan and derivatives such as chitooligosaccharides (CHOS) originating from the fungal cell wall [25]. The potential of these has challenged researchers to develop chemical analogs with similar characteristics or distinct from known natural inducers of the plant’s immunity [26,27]. However, despite intensive research, success in field trials using BCAs is very limited due to variations in ecological parameters like plant physiological and genetical status, climatological conditions, etc., which increase the variability of the desired BCA effect [28,29,30]. Therefore, their use is more restricted to the cultivation of greenhouse crops, where environmental conditions are more controllable [31]. In the field, a more reliable disease control could rely on combinations of BCAs and fungicides. As these types of combinations could reduce the fungicide dose (under the MRLs) or the frequency of application and improve disease control, they translate the principles of IPM into practice. In addition, such a strategy of combining antifungal treatments with different modes of action would fit within the advice of the fungicide resistance action committee to reduce the selection pressure on pathogens and thereby the chances of resistance development [32]. 

In this review, combinations of either antagonists or inducers of resistance with fungicides against fungal plant pathogens will be discussed. As mentioned before, these antagonists and inducers of resistance affect plant pathogens in a direct or indirect way, respectively, which influences the way in which disease control can be applied. Moreover, they can be of biological or chemical origin, and this information is also necessary for registration. Therefore, these types of combination treatments will be discussed separately. Some BCAs, however, can act in both a direct and indirect way. For example, the well-known *Trichoderma* spp. are plant symbionts that colonize plant roots and improve nutrient uptake. Besides their well-reported antagonistic effects on plant pathogens, some *Trichoderma* spp. are also strong inducers of plant defense mechanisms [24]. 

## 2. Biological BCAs

To inhibit fungal pathogens, fungicides have been developed that target different components or mechanisms of the fungal cell, including respiration, nucleic acid metabolism, cell membrane integrity, protein synthesis, signal transduction and cell mitosis [32]. However, some fungicides perform these activities without distinguishing between harmful pathogens and nontarget organisms such as beneficial micro-organisms in soil and living BCAs [33]. As such, fungicides could impact the growth of BCAs or reduce their population size, making the biocontrol treatment ineffective. Therefore, knowledge of compatibility of fungicides and BCAs is crucial to allow combined applications. For example, by using bacterial BCAs, the biocontrol effect could be less impacted by fungicides acting on more fungi-specific targets [34]. Alternatively, fungal BCAs can be used that are selected or developed for enhanced resistance to specific fungicides [35]. Each combination of a BCA and a fungicide should therefore always be examined. Usually, the inherent resistance of a BCA against a fungicide is first examined in vitro [36]. Combinations of such resistant BCAs and the corresponding fungicide can subsequently be confirmed in vivo and further fine-tuned for optimized disease management capacity.

### 2.1. Combinations of Fungicides with Biological Antagonists

BCAs can manage plant diseases through direct antagonistic effects on plant pathogens via parasitism, antibiosis or competition for nutrients or space. Parasitism is a relationship between two organisms in which one directly gains nutrients from the other. A specific type of parasitism well-known in the biocontrol field is mycoparasitism, in which fungal plant pathogens are parasitized by biocontrol fungi, reportedly often *Trichoderma* spp. [37,38]. The second mechanism, antibiosis, takes place between two organisms when one produces antimicrobial metabolites that directly impact the growth or metabolism of the other organism. These antimicrobial products are produced at very low concentrations, and they are only locally distributed and have a short life-span; therefore, their toxicological risks to humans are low [19]. Finally, competition occurs when two organisms require the same limited nutrients or space. These protective mechanisms of direct BCAs are often complex and rely on different and multifaceted modes of action, which is expected to lower the chances of resistance development. Another advantage of direct BCAs includes the possibility to investigate inhibitory effects via simple bioassays between only the antagonist and the pathogen, which is more straightforward than investigating indirect effects between the pathogen, inducer of resistance and the plant [20]. Despite this, complete disease control can only be obtained when BCAs are combined with fungicides [39]. In the following paragraphs, such combinations will be described but no distinction will be made between competition, antagonism and parasitism since the main mechanism of control is not always clear; instead, a distinction will be made based on their time (either pre- or post-harvest) or method of application or origin [19]. 

Microbial antagonists with a direct action have reportedly been combined with fungicides to control post-harvest diseases. An advantage of the post-harvest application of antagonists and fungicides includes the simple treatment via dipping of harvested fruits in one solution. However, as mentioned before, the application in a mixture implies that the antagonist is inherently resistant against certain fungicides. As such, improved control of ber fruit rot (caused by *Alternaria alternata*) was obtained when harvested fruits were dipped in a mixture of fungicide-resistant *Trichoderma* spp. and various systemic and non-systemic fungicides at low doses of 50 or 100 ppm [40], as compared to the 10-times higher doses typically applied for fungicides on ber fruit [41]. Some *Trichoderma* isolates caused a latent infection which was completely suppressed with the combination. In a different study, on stored apples, a mixture of the biocontrol yeast *Cryptococcus laurentii* and thiabendazole, at 10% of the standard dose, resulted in the highest and longest control of another important post-harvest pathogen, *B. cinerea* [42]. The combination was even more effective against a thiabendazole-resistant isolate of *B. cinerea*, also providing longer disease control compared to treatment with the biocontrol yeast alone. Therefore, BCA-fungicide combinations could have potential against populations of fungicide-sensitive and fungicide-resistant populations, which are becoming more and more prevalent [43]. Similarly, on harvested apples but using newer fungicides, a solution of the biocontrol yeasts (*Rhodosporidium kratochvilovae* or *C. laurentii*) with a low dose of either boscalid or cyprodinil was more effective against blue mold caused by *Penicillium expansum* than the treatment by itself [44]. Interestingly, lower fungicide residues were observed with the combination treatment even when compared to single treatment with the same fungicide at the same low dose. Most successful post-harvest treatments involve the combined application of biocontrol yeasts and fungicides, which is likely due to the ability of yeasts to tolerate extreme environmental conditions, making them appealing for food application. As such, yeasts can survive in routinely used storage conditions, including low oxygen levels, low temperatures and UV radiation, but also in conditions specific to foods, such as low pH and high sugar concentrations [45]. Sometimes, the pre-harvest application of fungicides is more efficient against post-harvest pathogens, but also this approach could be improved via combinations with BCAs. As such, the combined pre-harvest application of *Epicoccum nigrum* and various fungicides could reduce the fungicide dose three-fold without affecting the management of brown rot (caused by *Monilinia* spp.) on harvested peaches during four different field trials [46]. Disease reduction was most effective in years with lower disease severity. 

Fungal antagonists also improve disease control when combined with traditional fungicides against pre-harvest pathogens. In particular, the application of *Trichoderma* spp. against soilborne pathogens is known for these reasons. *Trichoderma* spp. are inherently resistant against some fungicides, allowing the combined application in a mixture. Such a combination of *T. virens* and thiophanate-methyl was found to be compatible and more effective than either treatment alone against *Fusarium solani* and *Fusarium oxysporum* in field trials of dry bean production [47]. The vegetative growth of the plants and yield was also significantly increased for the combination compared to single treatment. Similarly, the combined application of *Trichoderma* spp. with a low dose of fluazinam was found to be more effective to control avocado white rot (caused by *Rosellinia necatrix*) than either treatment alone [48]. Finally, though *Trichoderma* spp. were found not to be effective against *F. oxysporum* and *Acremonium strictum* in an in vitro setting, combining them with a low dose of the broad-spectrum fungicide tolclofos-methyl was superior to the fungicide only [49]. 

The rhizosphere of plants forms a source of bacteria not only important for plant resistance but also for direct biocontrol in pre-harvest applications. Similar to fungal antagonists, these bacterial antagonists mainly improve disease control against soilborne pathogens. For example, the incomplete disease control of *Bacillus megaterium* against *F. oxysporum* on tomato could be improved when combined with a low dose of the fungicide carbendazim in plant-packs [50]. The combination provided full disease control, even outperforming application with the fungicide at a 10-fold higher dose. Similarly, in the same set-up, combined application of rhizobacteria *P. fluorescens* and a 10-fold reduced dose of benomyl was more effective than treatment with either alone and reduced the disease as much as a full dose of the fungicide alone [51]. Interestingly, some biological antagonists can survive on the leaves of plants, which allows spray application against foliar pathogens. As such, *Bacillus subtilis* is a rhizobacterium that has been widely tested for its production of antibiotics that affect the cell wall of plant pathogens [52]. In multiple greenhouse trials, the foliar application of *B. subtilis* with azoxystrobin provided the highest yield and the best disease control against powdery mildew (caused by *Podosphaera xanthii*) on zucchini, compared to both treatments alone [53]. 

Fungicides have also been combined with multiple fungal and bacterial BCAs to enhance their disease management capacity pre-harvest. A combination of *P. fluorescens*, *Mesorhizobium cicero* and *T. harzianum* with the fungicide Vitavax^®^ (active ingredients: carboxin and thiram; Haryana, India) provided the highest seed germination, grain yield and the lowest wilt incidence (caused by *F. oxysporum*) in pot and field experiments of chickpea [54]. Moreover, in field experiments of rice, the combination of *T. harzianum*, *P. fluorescens* and carbendazim was more effective against *Magnaporthe oryzae* in comparison to their individual application [55].

Derivatives of living organisms like plant extracts are also known as direct BCAs that can be combined with fungicides. Synergy was observed between either CHOS or chitosan and various synthetic fungicides on strawberry flowers [56]. The combination of the fungicide at a 100-fold reduced dose and chitosan or CHOS yielded a protection level against *B. cinerea* similar to the fungicide at full dose. A similar combination with a 10-fold reduction of the fungicide dithianon was more effective in controlling scab (caused by *Venturia inaequalis*) than the fungicide alone at the recommended dose in field trials of apple [56]. Moreover, plant extracts of *Inula viscosa* combined with the fungicide iprodione at a reduced rate were as effective against *B. cinerea* on bean plants as the full dose of fungicide [57]. 

### 2.2. Combinations of Fungicides with Biological Inducers of Resistance

Various biotic and abiotic stresses are well known to regulate the natural plant defense mechanisms by triggering induced resistance, which can be defined as an enhanced physiological state of defense that prepares plants against future pathogenic attacks. There are two main reported types of induced resistance: systemic acquired resistance (SAR) and induced systemic resistance (ISR). Both provide long-lasting resistance against plant pathogens but differ in the signaling molecules and pathways that result in such an increased state of alertness [58]. As such, the induction of SAR is usually activated by pathogen infection and requires the signaling molecule salicylic acid (SA) to accumulate pathogenesis-related proteins [59]. In contrast, ISR is triggered by beneficial micro-organisms and usually does not rely on SA but is dependent on pathways regulated by jasmonate and ethylene [60]. Moreover, there are other types of interactions between biological BCAs and plants (such as symbiosis) that can induce the defense mechanisms of plants. For example, endophytic fungi have been shown to colonize banana plants and thereby induce systemic resistance against *Radopholus similis* [61]. However, these types of symbiotic interactions fall outside the scope of this review. 

The amount and variety of mechanisms involved, and the absence of a direct interaction with the pathogen, implies that there is limited selection pressure on pathogens [62]. It is therefore unlikely that resistance develops against inducers of plant resistance. In addition, as these inducers activate the plant defense response that produces molecules which are generally present in natural environments for the communication between plants and micro-organisms, it is assumed that the induction of resistance poses very low toxicological and ecological risks to nontarget, beneficial organisms and humans [19]. Despite this, induced resistance often provides only 20–85% disease control and is thus rarely complete [63]. Moreover, the unpredictability of disease control due to environmental variations in crop nutrition, genotype and disease severity has raised further concerns [64]. To maximize efficiency and allow commercial application, they can be combined with fungicides. In the first instance, it is expected that, in such combinations, the systemic effect of inducers of resistance improves the disease control of non-systemic fungicides that only provide local disease control at the site of application. However, such an improved effect on disease control is also true for systemic fungicides, which do translocate through the plant but rarely move to all plant parts [51,65]. Another advantage of systemic inducers of resistance includes the possibility to apply the BCA as a seed treatment or on the roots of plants against foliar pathogens. Therefore, these types of combinations are more regularly effective against leaf spots, blights and mildews. Moreover, the chances of resistance development decrease by using these types of combinations [66]. 

#### 2.2.1. Combining Fungal Inducers of Resistance and Fungicides

*Trichoderma* spp. are beneficial fungi in the rhizosphere of plants of which some species are reported to act as BCAs either by directly antagonizing other pathogens or indirectly by inducing ISR [67]. When applied in alteration with a fungicide, the latter does not impact the growth of the BCA, and disease control performance is enhanced. In corn, for example, the spray application of difenoconazole-propiconazole followed by *Trichoderma harzianum* SH2303 was as effective in reducing southern corn leaf blight caused by *Cochliobolus heterostrophus* as a sequential spray with the fungicides, while the BCA alone was not effective [68]. Thereby, the combination allows a two-fold reduction of the fungicide dose to control southern corn leaf blight. Similarly, alteration of *T. harzianum* with dicarboximide fungicides was found to be as effective against grey mold (caused by *Botrytis cinerea*) on tomato plants as single dicarboximide treatment, while treatment with *Trichoderma* alone resulted in variable disease control [69]. Nevertheless, the combination allows a two-fold reduction in the number of fungicide sprays.

Even in combination with fungicides, the use of biological inducers of resistance can result in variable disease control, as their mode of action might be independent and just additive, or more variable results might be related to the lowered dose rate of the fungicide. A combination of carbendazim and *Trichoderma* sp. Tri-1 could reduce the fungicide dose by 25–50% to control *Sclerotinia sclerotiorum* on oilseed rape [70]. The highest disease control was obtained in fields where a rice–oilseed rotation was used, which generally is associated with lower disease occurrence. In the same way, alteration of chitosan with the fungicide chlorothalonil against *Didymella bryoniae* was found to be as effective as a continuous spray with the fungicide during a field trial on watermelon when disease severity was low [71]. However, during another field trial with high disease severity, alteration with chitosan was found to be ineffective against *D. bryoniae*. Therefore, sufficient research must be performed in different environmental conditions to reveal the true potential of combining a biological inducer of resistance with a fungicide. 

When applied as a mixture, the compatibility of *Trichoderma* spp. with fungicides needs to be examined [72]. If the fungicide impacts the survival of the BCA and the compounds cannot be administered separately, alternatives need to be explored. The combination of *Trichoderma* spp. and the fungicide iprodione against *S. sclerotiorum* required the selection of iprodione-resistant isolates of *Trichoderma* spp. [35]. Soil application of iprodione-resistant *Trichoderma virens* combined with iprodione resulted in a synergistic interaction and managed disease most effectively on cucumber. Alternatively, the administration of fungicide-sensitive *Trichoderma* spp. can be physically separated from the fungicide to allow such a combination. The main diseases threatening cotton seedlings are pre-emergence damping-off by *Pythium* spp. and *Rhizopus oryzae* and post-emergence damping-off by *Rhizoctonia solani* [73]. *Trichoderma* spp. can effectively manage pre-emergence damping-off via the local induction of phytoalexins in the cotton root. However, it cannot access the hypocotyl and therefore there is no phytoalexin production in this part of the plant, leaving it unprotected from post-emergence damping-off [74]. On the contrary, fungicide seed treatments can control post-emergence pathogens but they are not effective against pre-emergence pathogens. Hence, a combinatorial seed treatment, in which the fungicide chloroneb is first applied, followed by a coating of the seeds with a latex sticker after which *Trichoderma* spp. are applied to the seeds, manages both phases of damping-off effectively. 

#### 2.2.2. Combining Bacterial Inducers of Resistance and Fungicides

In addition to fungal inducers of resistance, such as some *Trichoderma* spp., also bacterial ones have been successfully used in combination with fungicides. Maneb and mancozeb are non-systemic fungicides that need to be repeatedly applied to manage fungal plant diseases as they are only effective on contact. When roots of corn were drenched in a solution containing *Bacillus cereus* C1L, a rhizobacterium known to induce plant resistance, the number of leaf sprays with maneb which was necessary to control southern corn leaf blight (caused by *C. heterostrophus)* could be reduced two-fold [75]. In addition, while treatment with the fungicides alone negatively affected plant growth, the combinatorial treatment significantly increased plant growth, as compared to untreated plants under natural conditions. Similarly, seed treatment of B. cereus, which induces systemic resistance, could reduce the number of sprays of another non-systemic fungicide, chlorothalonil, to manage early blight (caused by *Alternaria solani*) in tomato [76]. The frequency of fungicide sprays therefore could be scaled down from 20 to 10 applications while the yield was unaffected over a 90-day field study, confirming the long-lasting effect of inducers of resistance on plant defense mechanisms. Combinations of bacterial inducers of resistance and systemic fungicides are also relevant against various diseases on crops with economic importance. For example, leaf spots caused by *Cercospora beticola* reduce the harvest of sugar beets. In the past, the disease was controlled by fungicides such as triphenyltin hydroxide, benomyl and thiophanate-methyl; however, the pathogen has become resistant. The biocontrol agent *Bacillus mycoides* was able to induce plant resistance and thereby reduce Cercospora leaf spot by 38–91% in six different field trials [77]. The addition of an alternative fungicide, propiconazole, in a mixture with *B. mycoides* reduced the variability and always allowed effective disease management while lowering the chances of resistance development. Similarly, in field trials with different wheat varieties, ranging from susceptible to moderately resistant, the combination of *Lysobacter enzymogenes* strain C3, known to induce disease resistance in the plant, and the fungicide tebuconazole was consistently effective against Fusarium head blight (caused by *Fusarium graminearum*) while disease control with the fungicide or BCA alone was variable [78]. Indeed, while in half of the field trials, application of the BCA alone was effective on susceptible varieties but not on moderately resistant varieties, in the other field trials, it was either not effective on any variety or effective on all. As mentioned before, such variable differences in the disease control activity of a BCA are assumed to be dependent on environmental conditions. Interestingly, the combined treatment of BCA and tebuconazole in these field trials did not show this high variability and was always effective [78]. Finally, these types of combinations are also reported to be relevant against powdery mildews and fruit rots. When applying the biocontrol bacterium *Pseudomonas fluorescens* to treat powdery mildew and fruit rot (caused by *Leveillula taurica* and *Colletotrichum capsici,* respectively) in chili cultivation in the field, disease control was found to be incomplete [79]. However, combined application of the BCA with a two-fold reduced dose of the standard fungicide, azoxystrobin, was as effective as the fungicide at standard dose. 

### 2.3. Conclusions

To conclude, fungal disease control can be improved when fungicide-compatible BCAs are combined with fungicides. These treatments may have the potential to develop new antifungal strategies for integrated pest management since the chances of resistance development are lower and the fungicide dose might be reduced compared to traditional treatment with single fungicides. If the fungicide impacts the growth and development of the BCA, they should be separated in time or space from fungicides, which is evidently not possible for most direct BCAs that are applied together with fungicides. Advantages of indirect inducers of resistance include the long-lasting, systemic disease control. As a result, the application will protect the entire plant, even parts that are hard to reach by spray applications. It seems that the potential of these types of combinations has not been completely explored, since, to the best of our knowledge, there are no reports on the combined use of biological inducers of resistance and fungicides in post-harvest applications. However, it is reported that biological inducers of resistance do provide incomplete disease control in this setting [80,81]. In contrast to BCAs directly antagonizing pathogens, which have been used since the 1980s, these ISR-inducing BCAs have been more recently developed for treatment in post-harvest disease control, which could explain the research gap [39]. However, since these combinations of fungicides and ISR-inducing BCAs might also be valuable against post-harvest diseases, they should be included in future research. Nevertheless, disease control through the use of BCAs might remain variable, even in combination with fungicides, and therefore such combinations need to be fully investigated under natural conditions [70,71]. 

## 3. Chemical BCAs

### 3.1. Combinations of Fungicides with GRAS Antagonists

Chemicals recognized as generally regarded as safe (GRAS) by the United States Food and Drug Administration (FDA) are considered as non-toxic since extensive historical use did not cause any health hazards. Salts like sodium benzoate, sodium methylparaben, sodium bicarbonate and potassium sorbate are considered as GRAS and their antifungal activity can be used as part of IPM post-harvest. However, GRAS compounds are usually not considered as BCAs. Despite this, they generally impact fungal cell membrane permeability and fungal nutrient transport but are nontoxic to humans and can be used as part of organic farming [82,83]. As the application of these compounds has similar characteristics to that of BCAs and also has some additional advantages, including low cost and ease of handling, they will also be discussed in this review. For example, dipping citrus fruits in the salts sodium benzoate or sodium methylparaben was very effective against sour rot (caused by *Geotrichum citri-aurantii*) [84]. Moreover, GRAS compounds have been successfully used in edible coatings on harvested fruits [82]. For instance, such coatings consisting of potassium sorbate, sodium benzoate or potassium silicate on mandarins are able to reduce citrus anthracnose (caused by *Colletotrichum gloeosporioides*) severity up to 70% [85]. However, despite some success, their disease control potential is usually highly dependent on the plant cultivar and often shows limited fungicidal activity. Therefore, their usage should be integrated with other disease management strategies to allow commercial application and reduce the fungicide dose or application frequency. In this review, the combined use of GRAS antagonists and fungicides will be further discussed. As such, the addition of sodium bicarbonate to imazalil importantly improved its activity against both fungicide-sensitive and fungicide-resistant isolates of *Penicillium digitatum* on harvested lemons [86]. Similarly, Kanetis et al. observed that the efficiency of fungicides such as azoxystrobin, fludioxonil or pyrimethanil is significantly increased when mixed with GRAS sanitizers such as sodium bicarbonate when treating citrus green mold caused *P. digitatum* [87]. However, the disposal of sodium bicarbonate creates environmental problems because of its pH and its electrical conductivity. Alternatively, potassium sorbate is another common food additive that can enhance the effectiveness of various post-harvest fungicides against citrus green mold and sour rot under commercial conditions [88]. The combinations were even effective against fungicide-resistant isolates of *P. digitatum*. 

Combinations of GRAS compounds and fungicides could also be valuable pre-harvest. As such, disease control in tomato plants naturally infected with *B. cinerea* was drastically improved when a peroxyacetic acid-based sanitizer was combined with the fungicide SWITCH^®^ (active ingredients: cyprodinil and fludioxonil; Syngenta, Switzerland) [89]. The combination allowed a 75% reduction in the standard fungicide dose while effectively suppressing grey mold. Moreover, the combination of GRAS-classified zinc oxide nanoparticles with either mancozeb, carbendazim or thiram was more effective than either compound alone against various pathogenic fungi, *Penicilium expansum* being the most sensitive [90]. As a result, these mixtures could improve disease management and reduce the fungicide dose and thereby the residues on harvested fruits.

### 3.2. Combinations of Fungicides with Chemical Inducers of Resistance

The enormous potential of biological inducers of resistance reported to protect plants against diseases has challenged researchers to develop chemical analogs of these with similar systemic disease control and low risk of pathogen resistance development [26]. The focus shifted mainly towards compounds that are involved in the first phase of induced resistance and includes the commercially available SAR-activator S-methylbenzo (1,2,3) thiadiazole-7-carbothiate (ASM), a synthetic analog of SA [91]. The combination of ASM with fungicides can be applied as one treatment or in alteration and has been extensively studied in various crops against different pathogens, as exemplified below. As these activators result in the production of signaling molecules that are already universally produced by micro-organisms for their communication with plants, they are also not expected to hinder the environment, similar to biological inducers of resistance.

#### 3.2.1. Mixtures of ASM and Fungicides

When ASM is applied in combination with fungicides, they can be mixed in a spray tank, commonly referred to as tank-mixtures [32]. Tank-mixing ASM with fungicides is reported to improve disease control against various fungi. Some interesting examples of such combinations are discussed in this paragraph; additional examples are presented in Table 1. The addition of ASM has been shown to increase the efficiency of strobulin fungicides, which are classified as “reduced risk” fungicides by the United States Environmental Protection Agency (US EPA) due to their low toxicity to human health and the environment, against fungal pathogens [92]. Field applications on the leaves of spinach demonstrate the potential of an alternative combination between ASM and strobilurin against white rust (caused by *Albugo occidentalis*) [93]. Leaf quality was also improved when the standard fungicides, mefenoxam plus copper hydroxide or trifloxystrobin, were combined with ASM. Similarly, the disease management of scab (caused by *Cladosporium oxysporum*) on passionfruit was significantly enhanced when ASM was combined with the industry standard fungicides or azoxystobin, as an alternative combination [94]. The incorporation of ASM also improved the amount of marketable fruit produced, compared to the standard program. However, the addition of ASM did not affect fruit spots caused by *A. alternata*, indicating that ASM is not effective against all pathogens that infect passionfruit. The combination of ASM and fungicides can provide long-term disease control and, therefore, allow treatment of corms of gladiolus before field planting [95]. As such, treatment with either ASM or azoxystrobin alone was not effective against corm rot caused by *F. oxysporum*, but when the two were applied as a mixture, they provided long-lasting disease suppression during the entire season on corms of gladiolus [95]. Disease control against blue mold (caused by *Peronospora tabacina*) on tobacco consists of the fungicides dimethomorph plus mancozeb alternated with azoxystrobin. The latter two non-systemic fungicides require repeated application on the entire plant, grown at high densities inside shade tents, making the application very laborious [96]. The fungicide program could be greatly improved when the fungicides were combined with ASM, resulting in higher yields and improved (systemic) disease control [97]. 

To conclude, mixtures of ASM and fungicides might be a useful strategy to broaden or intensify disease control in different applications and tissues, as exemplified above, and decrease resistance development compared to applying fungicides alone.

#### 3.2.2. Mixtures of BABA and Fungicides

β-aminobutyric acid (BABA) rarely occurs naturally in plants but also induces broad-spectrum resistance against plant pathogens and has also been used in combination with fungicides [102]. The combination of BABA and the fungicide folpet (N-(trichloromethylthio)phthalimide) or BABA and fosetyl-aluminum in a tank-mixture effectively controlled downy mildew, caused by *Plasmopara viticola*, in field grown grapevines at a two-fold reduction of the standard fungicide doses [103]. Similarly, additive effects between BABA and fluazinam were observed against late blight infection (caused by *Phytophthora infestans*) in potato field trials. The combination allowed a 25% reduction of the fungicide dose to effectively control late blight infections [104].

#### 3.2.3. Alternation of Chemical Inducers of Resistance and Fungicides

Alternating the application of chemical inducers of resistance with fungicides is a different strategy to manage plant diseases effectively while reducing fungicide doses and the risk of pathogen resistance. To control late blight, caused by *P. infestans*, tomato plants are repeatedly sprayed with phosphorous acid. Alteration of SA with the fungicide reduces the fungicide dose two-fold and manages late blight as effectively as continuous application of phosphorous acid [105]. The effect of multiple inducers of resistance on the infection of spring barley by *Rhynchosporium commune* and *Blumeria graminis* was investigated by Walters et al. [106]. The combination of SA, BABA and cis-jasmone could efficiently manage disease under glasshouse conditions. However, disease control by the three elicitors was very variable by year and variety and never complete during field experiments over three years. Disease control by alternating the three elicitors with fungicides was always equal to—and, in some cases, better than—application with the fungicides alone. Similarly, disease control of downy mildew (caused by *Peronospora belbahrii*) on basil plants by a foliar spray of ASM or BABA could be significantly improved when combined with a fungicide mixture of potassium phosphite and azoxystrobin at one time post-inoculation [107]. The efficiency of such a combination was similar to the standard treatment of downy mildew, consisting of the fungicide mixture applied three times every week. Alteration with BABA or ASM could therefore reduce the application of these fungicides three-fold. Combinations with chemical inducers of resistance can also be useful to control post-harvest diseases. Guazatine is effective against *Fusarium* spp. but less so against *Alternaria* spp. and *Rhizopus* spp., which are among the most common post-harvest pathogens on rock melon [108]. A pre-harvest foliar spray of ASM combined with a fruit dip with guazatine post-harvest was effective against *Alternaria* spp. and *Rhizopus* spp. and was generally the most effective to decrease disease in stored rock melons [108].

### 3.3. Conclusions

To conclude, combinations of fungicides and chemical inducers of resistance or GRAS antagonists have been clearly demonstrated to improve disease control similarly to combinations with biological BCAs. Such combinations have been demonstrated to be effective against resistant plant pathogens and can be effective in both pre-harvest and post-harvest settings [86,88]. Similar to biological inducers of resistance, also chemical inducers of resistance can induce long-lasting systemic disease control that protects the entire plant. However, as compared to biological inducers, they have the additional advantage of being (more) compatible with fungicides. The latter is also true for antagonists classified as GRAS which directly impact plant pathogens. However, the application of both in combination with fungicides can differ. Indeed, while GRAS antagonists are mainly applied by mixing with fungicides, inducers of resistance are commonly applied either in a mixture or in alteration with fungicides. Finally, these types of combinations are mainly used to improve the activity of fungicides or reduce their dosing and/or to decrease the chances of pathogen resistance development while ensuring low toxicity to humans and the environment.

## 4. Future Perspectives

As the impact of the overuse of synthetic fungicides will become increasingly apparent due to the rising threat of resistant pathogens and the deleterious effects on soil productivity and human and animal health, alternatives need to be explored [7,9,12,109]. In this regard, research has focused on BCAs, as their toxicological risks to humans and the environment are low. As such, disease control by BCAs results in highly regulated processes between pathogens and plants that involve multiple metabolites. These processes are ubiquitous in natural environments and humans and other organisms have been exposed to them for years while negative effects are unknown [19,94,104,110] However, their disease control activity is often found to be incomplete and highly dependent on environmental conditions [39,63,111,112]. Therefore, integrated pest management approaches consisting of combinations of either systemic or non-systemic fungicides with antagonists or inducers of resistance are recommended [113].

Fungicides with a non-systemic mode of action are usually active against a broad spectrum of plant pathogens as they interfere aspecifically with different metabolic processes. It is commonly believed that the risk of fungicide resistance development for such compounds is low since this would require multiple mutations in the genome of the pathogen [43]. Despite this, resistance is observed and these fungicides often have inferior activity against pathogens compared to specific fungicides, causing the need for frequent application at high dose. For example, resistance of *B. squamosa* to both iprodine and vinclozolin has been reported [114]. Generally, managing the pathogen requires weekly sprays of fungicides at high doses, between 1 and 2 kg/ha. Hence, combining them with direct antagonists or inducers of resistance can enhance the disease control capacity of these fungicides and thereby respond to the increasing restrictions on the use of non-systemic fungicides [32]. An additional advantage of such a combination with inducers of resistance includes the systemic disease control. As a result, the combination will protect the entire plant, even parts that are hard to reach. Furthermore, the addition of such additional BCA will decrease the chances of resistance development even more.

Strobulin fungicides are a class of systemic fungicides that are classified as “reduced risk” by US EPA due to their low toxicity to human health and the environment [92]. However, these systemic fungicides target one or few sites in biochemical pathways and are therefore at high risk for resistance development by pathogenic fungi. Similarly, resistance of pathogens is often described against benzimidazoles, triazoles and dicarboximides, which are not labeled as “reduced risk” by US EPA. Combining them with antagonists or inducers of resistance through tank-mixing or alternated applications could not only improve disease control but also reduce the risk of resistance development. If resistance is already present, these types of combinations have also proven to be effective [42,86,88].

The characteristics of BCAs are dependent on their origin, either chemical or biological, and their mode of action, either direct or indirect (Table 2). For example, screening for BCAs that induce resistance requires more complicated assays on plants than screening for direct BCAs, which is usually possible in vitro [20]. Moreover, BCAs that enhance the activity of fungicides should be compatible. When they are applied together, the fungicide does not only interact with pathogens but also with BCAs. As BCAs are designed to either improve plants’ defense mechanisms or directly impact plant pathogens, it would be unlikely that such combinations negatively affect the activity of fungicides. Since chemical BCAs are non-living, they are therefore not impacted by synthetic fungicides unless physical incompatibility occurs. However, the risk that fungicides would negatively impact the growth or survival of living BCAs is considered to be much higher. In this case, they could be separated in time or space when the main antifungal effects are indirect. However, as some BCAs impose direct antagonistic effects on pathogens, such a separation is usually not possible. Therefore, future research should focus on optimizing BCAs and their timing of application to tolerate certain fungicides. Another advantage of chemical BCAs includes their stability and long shelf-life, while living BCAs often require refrigeration to maintain their cell viability, increasing the total cost [115]. Despite this, chemical BCAs are still synthetic compounds and, even if they only exert indirect effects against plant pathogens and therefore do not affect human health, they could still be identified as dangerous by consumers in our chemophobic society [116]. Therefore, society should be informed and educated on these compounds to make sure that the best plant protection methods could be applied. Combining fungicides with BCAs is one of the most immediate methods that can be integrated on a commercial scale by farmers to decrease the residues of fungicides and hence decrease environmental and human health risks.

In many cases, combinations allow reduced rates and/or applications of fungicides to manage plant diseases of both sensitive and resistant populations. These benefits will directly reduce food losses and thereby production costs, which will stimulate farmers to change from fungicide-based agriculture to an integrated approach. Some European countries are increasingly encouraging farmers to use alternative plant protection methods via indirect economic benefits through taxes and subsidies [118]. Moreover, these types of combinations provide an answer to the consumer’s demand for a more sustainable agriculture and could be a tool for achieving reduced or even zero-residue produce. In addition, the legislation, requiring IPM and constricting more and more active ingredients, places increasing pressure on farmers to reduce their pesticide usage to reduce residues on their products and to protect the environment and human health [16]. Despite this, without adequate knowledge, it is difficult to break old habits and such a change might be easier for new farmers rather than established farmers. Therefore, free advisory services educating farmers on the benefits of IPM methods are valuable, especially when uncertainty about the efficiency of alternatives to pesticides overshadows [119].

## 5. Summary Points

Biocontrol agents (BCAs) can be of biological or chemical origin and can protect plants against pathogens indirectly, via the induction of resistance, or directly via parasitism, antibiosis or competition for nutrients or space. Due to uncontrollable environmental conditions, success in field trials is variable.Combinations of fungicides and BCAs improve disease control against a large variety of plant pathogens in a more reliable way.Combining antifungal treatments with different modes of action decreases the selection pressure on plant pathogens and reduces the chances of resistance development. If resistance is already present, combinations of fungicides and BCAs are reported to be effective.Antimicrobial products produced by both biological and chemical BCAs are nontoxic to humans and the environment. Combined with fungicides, the fungicide dose or application frequency necessary to control diseases can be reduced, contributing to a more sustainable agriculture and/or marketing concepts as low or zero-residue produce.Living BCAs need to be compatible with fungicides as the latter might negatively impact their growth. In the case of incompatibility, their method of application should be modified by separating them in time, via alternation, or space. Such separation is usually not possible for antagonists that directly affect plant pathogens. Therefore, research should also further focus on fungicide-resistant antagonists. Chemical BCAs are generally not affected by fungicides.Most common biological inducers of resistance that are combined with fungicides are *Trichoderma* spp. or *Bacillus* spp., whereas most research has focused on ASM as a chemical inducer of resistance. Advantages of these include the long-lasting, systemic effect which can greatly improve the application and disease spectrum of fungicides.Biological antagonists mainly originate from the rhizosphere and, as they impact plant pathogens directly, they are usually combined with fungicides against soilborne diseases.Public fear of chemicals is not necessarily based on real threats as chemicals recognized as GRAS and chemical inducers of resistance are nontoxic to humans and the environment. Advantages of these types of BCAs include the long shelf-life and stability, which are important benefits for supply chain and stock management by the suppliers.

## Figures and Tables

**Table 1 microorganisms-08-01930-t001:** Reported combinations of selected fungicides and S-methylbenzo (1,2,3) thiadiazole-7-carbothiate (ASM) as a way to improve disease control.

Pathogen	Fungicide	Crop	Result	Reference
*Thielaviopsis basicola*	Myclobutanil	Cotton	Best disease control	[98]
*Phytophthora capsici*	Mefenoxam, copper hydroxide and mandipropamid	Squash	Best disease control Highest yield	[99]
*P. xanthii*	Chlorothalonil	Cucumber	Best disease control Two-fold reduction of fungicide dose	[100]
*Plasmopara halstedii*	Oxathiapiprolin	Sunflower	Best disease control 10-fold reduction of fungicide dose	[101]

**Table 2 microorganisms-08-01930-t002:** Differences between the development, use and risks of BCAs based on their mode of disease control and their biological or chemical origin.

	Biological BCAs		Chemical BCAs	
	Inducers of Resistance	Antagonist	Inducers of Resistance	GRAS Antagonist
Chances of resistance [19,26,62]	Low	Low	Low	Dependent on MOA
Human health risks [19,110,117]	Low	Low	Low	Low
Ecotoxicological risks [19,94,104,110]	Low	Low	Low	Low
Screening for new BCA [20]	Assays on plants	Simple bioassays	Assays on plants	Simple bioassays
Plant protection [39,58]	Systemic	Local	Systemic	Local
Fungicide dosage [42,68,89,101]	Reduced	Reduced	Reduced	Reduced
Compatibility with fungicides [35,72]	Uncertain	Uncertain	Usually compatible	Usually compatible
Storage [115]	Specific conditions to maintain viability	Specific conditions tomaintainviability	Normal	Normal
Public opinion [116]	Positive	Positive	Negative	Positive

Abbreviations: BCAs, biocontrol agents; GRAS, generally regarded as safe; MOA, mode of action.

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
