# Peer review of "Combining Biocontrol Agents with Chemical Fungicides for Integrated Plant Fungal Disease Control"

_microorganisms, 2020, doi:10.3390/microorganisms8121930_

Round 1
Reviewer 1 Report
The review entitled Combining biocontrol agents with chemical 2 fungicides for integrated plant fungal disease control is very interesting and highlights, on the one hand, the problems inherent to the use of fungicides and the possible alternatives using either biological BCAs or chemical BCAs agents.
It would only be necessary to take into account a few small details
-Please Revise line 261-263
-Paragraph 302-314: I think it would be necessary to include some work done in Dr. Palou's group in reference to GRAS compounds used as an alternative to fungicides.
Soto-Muñoz et al., 2020. Curative activity of postharvest GRAS salt treatments to control citrus sour rot caused by Geotrichum citri-aurantii. DOI: 10.1016/j.ijfoodmicro.2020.108860.
Martínez-Blay et al, 2020. Edible Coatings Formulated with Antifungal GRAS Salts to Control Citrus Anthracnose Caused by Colletotrichum gloeosporioides and Preserve Postharvest Fruit Quality. DOI: 10.3390/coatings10080730.
Martínez-Blay et al, 2020. Control of major citrus postharvest diseases by sulfur-containing food additives. DOI: 10.1016/j.ijfoodmicro.2020.108713
Palou, L. 2018. Postharvest Treatments with GRAS Salts to Control Fresh Fruit Decay. DOI: 10.3390/horticulturae4040046.
In the final conclusions, it might be necessary to highlight that rather than increasing the doses of fungicides when they are no longer effective, it would be necessary to deepen and understand the mechanisms that allow the development of resistance. The alternative of combined use of BCAs with fungicides is good but it is incomplete because in many cases there is no microorganism that resists the doses of fungicide applied. In this case, the use of GRAS substances would be more promising as long as they do not interfere with health.
Finally, some type of graph or figure that summarizes the most relevant aspects of the review is missing. It could help to read the review and it would be very explanatory.
Reviewer 2 Report
In the manuscript 1009717 entitled “Combining biocontrol agents with chemical fungicides for integrated plant fungal disease control”, the authors discuss examples available in the current literature showing effects of combination of BCAs and synthetic fungicides against fungal plant pathogens.
This review is timely as well as necessary and shows us opportunities for integrated pest management. BCAs are often associated with organic agriculture but their use in IPM systems have often been neglected. The combination of BCAs with synthetic fungicides can be an additional tool in the IPM toolbox thereby helping to reduce the dosage of fungicides and opening new options for fungicide resistance management. BCAs can use different modes of action (MOA) such as competition, parasitism, antibiosis and the induction of resistance in crop plants. These MOA are not mutually exclusive meaning that some BCAs might use several MOA at the same time.
For the discussion of the different BCAs and their combination with synthetic fungicides, the BCAs were grouped into biological and chemical BCAs considering also aspects of registration. Furthermore, the biological BCAs were grouped into antagonists and biological inducers of resistance. Chemical BCAs were separated into GRAS antagonists and chemical inducers of resistance. In these groups the most recent examples of combinations of BCAs with synthetic fungicides were discussed. The presented examples are very diverse ranging from field to greenhouse applications in a vast array of different crop species. Moreover, examples of pre-harvest as well as post-harvest treatments are discussed showing the versatility as well as the necessity for pathogen-plant adapted applications strategies. In the “future perspectives”, aspects of combinations of systemic and non-systemic fungicides with BCAs are discussed as well as the perception of consumer of BCAs.
Overall, this manuscript is very well structured and prepared. The references also include the currently available literature. There are a few things which need further attention.
- Introduction
I would recommend to add a paragraph about MOA of synthetic fungicides and compatibility of BCAs with certain fungicidal substances due to “inherent resistance”. This would also support that combinations of BCAs with fungicides need to be evaluated in each case and that not all fungicides affect all fungi and fungal-like organisms in the same way. This is also a fact which is somewhat overlooked in the public discussion yet essential. Furthermore, it would be useful to introduce terms like fungicide-compatible etc. Please see also my comments below line122
- Future perspectives
Not all aspects of Table 1 are discussed within this section. A few sentences about ecotoxicological risks of BCAs would be very informative to the readers.
Specific comments
Line84 Do the authors also see the possibility to reduce the application frequency of certain fungicides?
Line122 Is it tolerance or more an inherent resistance meaning that the antagonist cannot be affected at all? Confusion with “developed resistance” should be avoided; The term “inherently resistant” is used later (Line 150). I think it would be good to introduce this term already here. An alternative would be to add a few sentences to the introduction about compatibility of BCAs with fungicides. Please also see my comment above.
Line177 Please change to F. oxysporum.
Lines261-263 Please remove italics where necessary
Line280 Leveillula taurica
Table 1 Storage: biological BCAs/antagonist: they also need specific storage conditions to maintain viability. For chemical BCAs/inducers of resistance the storage conditions are probably less specific. Please check this table again.
Line98 Delete Science of Total Environment
Reviewer 3 Report
Review report - Microorganisms-1009717
Combining biocontrol agents with chemical fungicides for integrated plant fungal disease control
General comments
This review reports on an integrated approach of combining biocontrol with fungicides. The subject matter of this article is very interesting and impotent. As written by the authors, this approach “reduces the fungicide doses to manage plant diseases and thereby their residue on harvested crops. In addition, such a strategy of combining antifungal treatments with different modes of action reduces the selection pressure on pathogens and thereby the chances of resistance development.” I agree.
However, I consider the manuscript has some weaknesses that should be carefully addressed.
Specific comments
Abstract
The abstract is too short. Please elaborate on the current status of this integration approach, and specify the challenges we face in the development and application required for the implementation of this integrated solution in commercial fields on a large scale.
Manuscript body
Lins 28-29 – The potential food losses due to diseases caused by pathogenic microorganisms and animal pests are estimated up to 38.2 % and 36.5 % of total yield losses in, for example, rice and potatoes, respectively.
It is unclear if “38.2 % and 36.5 %” are the distribution between microorganisms and animal pests or between rice and potatoes. Please rephrase the sentence.
Lines 39-40 – “food productivity has importantly increased over the last century [4].“ - Provide numbers – in what scale the use of fungicides increased over the previous century.
Lines 68-69 – “This broader term includes the application of living organisms and their derivatives to control plant diseases…” – Is the direct use of culture extract produced by Trichoderma spp. against phytopathogenic fungous is considered biocontrol treatment?
Line 125 – “low doses” – Please add a sentence or a paragraph that specifies the doses compared to regular doses used.
Line 190 - 2.2. Combinations of fungicides with biological inducers of resistance - Is it possible that the natural resistance of some cultivars (genotypes) to diseases are the result of symbiosis with beneficial fungi (endophytes)? If this idea could be true, perhaps it should be included.
Line 218 – “ Trichoderma spp. are beneficial fungi…” – this topic should receive a new sub-headline.
Lines 229-230 – “Even in combination with fungicides, the use of biological inducers of resistance can result in variable disease control. A combination of carbendazim and Trichoderma spp. …” – It should be noted that some Trichoderma spp. directly attack other fungi while others are endophytes and may induce resistance. It is wrong to include all Trichoderma spp. as inducers of resistance.
Line 256 – The bacterial part should receive a new sub-headline.
Line 263- “Similarly, seed treatment of B. Cereus…” Are these bacteria cause ASR or ISR? How the seed treatment produce a long-lasting effect that protects the adult plants? Please explain.
Line 274 – “different wheat varieties” – are these varieties differ in their susceptibility to the disease? Is there a different effect to the treatment in relation to the cultivars’ resistance/susceptibility?
Line 284 – perhaps it should receive a new sub-headline.
Line 293-294 – “… since to the best of our knowledge there are no reports on the combined use of biological 293 inducers of resistance and fungicides in post-harvest applications.” - this is not surprising since, in most cases, the use of fungicides in post-harvest crops is extremely limited because they might cause health hazards..." Please add an explanation.
Paragraph 3.2 is very long and hard to follow – split it into several sub-paragraphs and give each a sub-title.
The many examples in this paragraph can be organized in a table, enabling the shortening of the text.
Line 301 – Add an explanation of why Chemicals recognized as generally regarded as safe (GRAS) are considered BCAs and give cited examples. Are these compounds trigger the plant stress response? Why are thy Antagonists? Do you meant that they are pathogen-antagonist, act to protect against plant pathogens?
Lines 350-353 - “As such, treatment with either ASM or azoxystrobin alone was not effective against corm rot caused by F. oxysporum but when the two were applied as a mixture, they provided long-lasting disease suppression during the entire seasons [85].” - in what plant?
Line 354 – “Tank-mixing” – explain this term or replace it.
Line 402 – elaborate on the conclusion explanation and separate it into a new sub-paragraph and give it a sub-title.
Lines 433-434 – Are living BCAs can reduce the efficiency of fungicides? If this topic did not study, state that clearly.
Table 1 – explain all the abbreviations in the table footnotes.
Line 454-455 - “These benefits will directly reduce production costs .. “ – Why, are the biological and chemical BCAs that should be used has no cost? I guess you meant that they would reduce the cop losses – be more specific.
Also, the use of biological and chemical BCAs may contribute to crop marketing since these products will be more “green” - human and environmentally friendly and more organic.
Round 2
Reviewer 3 Report
Dear authors,
All the comments and suggestions have been addressed appropriately.
I appreciate the authors' corrections.
The current version of the manuscript has remarkably improved. It is much better than the previous one.
In my opinion, the manuscript is now suitable for publication.
Greetings!